# WHEN BIOLOGY HAS CHEMISTRY: SOLUBILITY AND DRUG SUBCATEGORY PREDICTION USING SMILES STRINGS

**Sarwan Ali, Prakash Chourasia & Murray Patterson** [*]
Department of Computer Science
Georgia State University
Atlanta, GA 30324, USA
{sali85,pchourasia1}@student.gsu.edu, mpatterson30@gsu.edu

## ABSTRACT

Drug discovery is a complex process that requires extensive research and development. One important aspect of drug discovery is the prediction of drug properties, such as solubility. In recent years, sequence-based embedding methods, such as SMILES strings, have gained popularity in the drug discovery community due to their ability to encode chemical structures. SMILES strings are text-based representations of chemical structures that can be easily processed by machine learning models. This research paper presents a study on predicting (i) the solubility ALOGPS (Ghose-Crippen-Viswanadhan octanol-water partition coefficient) and (ii) drug subcategories using traditional molecular fingerprints and sequence-based embedding methods (from the bioinformatics domain) of SMILES strings. The study investigates five types of embeddings: Morgan fingerprint, MACCS fingerprint, $k$-mers, and minimizer-based spectrum. Additionally, a weighted version of $k$-mers that employs inverse document frequency is used to assign weights to each $k$-mer within the spectrum. For the classification task (*i.e.*, drug subcategory prediction), we use the same embedding methods as input to several classifiers and report classification goodness using several evaluation metrics. For the regression task (*i.e.*, solubility ALOGPS prediction), we use several popular models *e.g.*, linear regression, and evaluate the performance using multiple evaluation metrics such as RMSE MAE, MSE, etc. The classification results indicate that the weighted $k$-mers method outperforms the baselines for predictive performance. The regression results indicate that the MACCS fingerprint with random forest regression model outperforms all other embedding methods and regression models. Overall, this study provides insights into the effectiveness of different embeddings, regression, and classification models for solubility and drug subcategory prediction, which can be helpful for future tasks such as drug discovery.

## 1 INTRODUCTION

Drug solubility is a crucial factor in drug discovery, as it affects the absorption and bioavailability of drugs in the human body. Therefore, predicting drug solubility and drug subcategories are essential steps in drug development. With the increasing availability of large-scale molecular data, machine-learning models have been developed to predict drug solubility. In recent years, there has been a growing interest in using molecular fingerprints Nakajima & Nemoto (2021); Keys (2005); Durant et al. (2002) and machine learning methods Francoeur & Koes (2021) with SMILES strings (see Figure 1 for example) to predict drug solubility.

In this study, we investigate five types of embeddings: Morgan fingerprint, MACCS fingerprint, $k$-mers Kang et al. (2022), weighted $k$-mers Öztürk et al. (2020), and minimizer-based spectrum.

**Remark 1** *Although fingerprints and $k$-mers-based embedding methods are already explored in the literature, there are two questions we are interested to answer in this paper (i) can we use the*

---

[*]Corresponding Author

Figure 1: Molecular structure for the drug named as S-Adenosylmethionine, with solubility ALOGPS value of 1.19e+00 g/l, and the following SMILES string:

```
C[S+](CC[C@H](N)C(O)=O)C[C@H]1O[C@H]([C@H](O)[C@@H]1O)N1C=NC2=C(N)N=CN=C12+
```

*existing methods for multiple tasks that involve both classification and regression at the same time? and (ii) Since k-mers is a popular method in the bioinformatics domain, can we use some alternate of k-mers from the bioinformatics domain, such as Minimizers Zheng et al. (2020); Edgar (2021) to generate the spectrum for SMILES strings and perform classification and regression analysis?*

The Morgan fingerprint is a circular fingerprint that encodes the presence of substructures within a molecule. The MACCS fingerprint is a binary fingerprint that encodes the presence or absence of predefined substructures. The $k$-mers and minimizer-based spectrum are sequence-based embeddings that encode the frequencies of overlapping sub-sequences of length $k$ and minimizers, respectively. The detail for embedding methods is provided in the appendix. We evaluate the performance of embeddings using various classification and regression models and show results using several evaluation metrics. The detail of the models and evaluation metrics is provided in the appendix.

We conduct our study on 6951 SMILES strings from the DrugBank dataset Shamay et al. (2018). For classification in Table 1, we can observe that the minimizers-based spectrum with KNN outperforms the other methods and classifiers for average accuracy, precision, recall, and weighted F1 score. For Macro F1 and ROC-AUC, weighted $k$-mers outperform the other methods. For regression analysis, our results (in Table 4 in appendix) show that the MACCS fingerprint with random forest regression model outperforms all other embedding methods and regression models. Overall, our study provides valuable insights into the effectiveness of embeddings, classification, and regression models for solubility and subcategory prediction. The findings can help researchers in drug discovery to choose the most suitable embedding and regression models for their specific applications.

| Embedding | Algo. | Acc. ↑ | Prec. ↑ | Recall ↑ | F1 (Weig.) ↑ | F1 (Macro) ↑ | ROC-AUC ↑ | Train Time (Sec.) ↓ |
|---|---|---|---|---|---|---|---|---|
| Morgan Fingerprint | SVM | 0.8838 | 0.8577 | 0.8838 | 0.8696 | 0.0591 | 0.5383 | 17.6993 |
| | NB | 0.8969 | 0.8454 | 0.8969 | 0.8697 | 0.0275 | 0.5068 | 3.5027 |
| | MLP | 0.8297 | 0.8493 | 0.8297 | 0.8390 | 0.0245 | 0.5239 | 17.4977 |
| | KNN | 0.9129 | 0.8543 | 0.9129 | 0.8798 | 0.0374 | 0.5130 | 0.2560 |
| | RF | 0.9109 | 0.8499 | 0.9109 | 0.8764 | 0.0258 | 0.5088 | 3.4253 |
| | LR | 0.9131 | 0.8520 | 0.9131 | 0.8784 | 0.0378 | 0.5148 | 2.8179 |
| | DT | 0.8569 | 0.8512 | 0.8569 | 0.8534 | 0.0333 | 0.5286 | 1.2680 |
| MACCS Fingerprint | SVM | 0.8705 | 0.8539 | 0.8705 | 0.8613 | 0.0520 | 0.5441 | 3.1812 |
| | NB | 0.2458 | 0.8473 | 0.2458 | 0.3698 | 0.0359 | 0.5224 | 0.5048 |
| | MLP | 0.8659 | 0.8444 | 0.8659 | 0.8547 | 0.0220 | 0.5175 | 21.0636 |
| | KNN | 0.9076 | 0.8447 | 0.9076 | 0.8741 | 0.0305 | 0.5107 | 0.0903 |
| | RF | 0.9057 | 0.8499 | 0.9057 | 0.8749 | 0.0344 | 0.5149 | 1.1254 |
| | LR | 0.9126 | 0.8331 | 0.9126 | 0.8710 | 0.0100 | 0.5000 | 3.2345 |
| | DT | 0.8227 | 0.8522 | 0.8227 | 0.8363 | 0.0457 | 0.5436 | **0.1100** |
| $k$-mers | SVM | 0.8190 | 0.8514 | 0.8190 | 0.8341 | 0.0413 | 0.5487 | 11640.03 |
| | NB | 0.7325 | 0.8425 | 0.7325 | 0.7816 | 0.0247 | 0.5149 | 2348.88 |
| | MLP | 0.8397 | 0.8465 | 0.8397 | 0.8426 | 0.0270 | 0.5311 | 7092.26 |
| | KNN | 0.9101 | 0.8480 | 0.9101 | 0.8766 | 0.0429 | 0.5167 | 68.50 |
| | RF | 0.9098 | 0.8449 | 0.9098 | 0.8740 | 0.0265 | 0.5075 | 655.47 |
| | LR | 0.8885 | 0.8423 | 0.8885 | 0.8642 | 0.0461 | 0.5286 | 1995.11 |
| | DT | 0.8429 | 0.8490 | 0.8429 | 0.8455 | 0.0397 | 0.5361 | 211.38 |
| Minimizers | SVM | 0.9128 | 0.8358 | 0.9128 | **0.8799** | 0.0108 | 0.5010 | 2205.95 |
| | NB | 0.0004 | 0.0003 | 0.0004 | 0.0003 | 0.0002 | 0.5071 | 4042.61 |
| | MLP | 0.9138 | 0.8357 | 0.9138 | 0.8726 | 0.0110 | 0.5029 | 1974.16 |
| | KNN | **0.9140** | **0.8579** | **0.9140** | 0.8724 | 0.0205 | 0.5166 | 102.87 |
| | RF | 0.9132 | 0.8351 | 0.9132 | 0.8721 | 0.0102 | 0.5044 | 6605.57 |
| | LR | 0.9131 | 0.8350 | 0.9131 | 0.8728 | 0.0109 | 0.5021 | 78.10 |
| | DT | 0.9130 | 0.8352 | 0.9130 | 0.8725 | 0.0107 | 0.5083 | 118.42 |
| Weighted $k$-mers | SVM | 0.8219 | 0.8355 | 0.8219 | 0.8368 | 0.0451 | **0.5490** | 9926.76 |
| | NB | 0.7490 | 0.8475 | 0.7490 | 0.7931 | 0.0360 | 0.5221 | 2564.96 |
| | MLP | 0.8288 | 0.8511 | 0.8288 | 0.8392 | 0.0270 | 0.5345 | 7306.79 |
| | KNN | 0.9122 | 0.8473 | 0.9122 | 0.8728 | 0.0307 | 0.5091 | 53.06 |
| | RF | 0.9135 | 0.8455 | 0.9135 | 0.8758 | 0.0245 | 0.5067 | 619.65 |
| | LR | 0.8928 | 0.8492 | 0.8928 | 0.8697 | **0.0595** | 0.5293 | 1788.37 |
| | DT | 0.8420 | 0.8518 | 0.8420 | 0.8461 | 0.0445 | 0.5347 | 147.47 |

Table 1: Average Classification results (of 5 runs) for different methods and datasets using different evaluation metrics. The best values are shown in bold.

## 2 URM Statement

Author S. Ali meets the URM criteria of the ICLR 2023 Tiny Papers Track.

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

APPENDIX

## A  RELATED WORK

Molecular fingerprints are widely used in cheminformatics to encode the structural information of a molecule as a binary vector Probst & Reymond (2018); Wigh et al. (2022). The fingerprints capture the presence or absence of certain substructures in the molecule and are useful for similarity searching, virtual screening, and predicting various molecular properties such as solubility, toxicity, and bioactivity. In recent years, there have been several studies on using molecular fingerprints and sequence-based embeddings to predict drug solubility Nakajima & Nemoto (2021). In a study by Cheng et al., the authors used a combination of molecular fingerprints and molecular descriptors to predict the solubility of 9471 compounds. They found that random forest regression and support vector regression performed better than other machine learning models Chen et al. (2018). Authors in Panapitiya et al. (2022) evaluated different Deep Learning Architectures for Aqueous Solubility Prediction. In a study by Rupp et al., the authors used a graph convolutional neural network to predict solubility based on molecular graphs. They achieved an R-squared value of 0.75 on a dataset of 1144 compounds Rupp et al. (2012). These studies demonstrate the potential of using machine learning models with molecular fingerprints and sequence-based embeddings for predicting drug solubility. However, there is still a need for further investigation into the effectiveness of different types of embeddings, classification, and regression models for solubility and drug subtype prediction.

## B  EMBEDDING METHODS

In this section, we explain the methods to generate numerical embeddings for SMILES strings, namely Morgan fingerprint, MACCS fingerprint, and $k$-mers-based spectrum. Then we discuss an embedding method, which we took from the biology literature, called minimizers, to generate the spectrum. Finally, we discuss a modified version of the $k$-mers spectrum by assigning weights to $k$-mers using the inverse document frequency (IDF) idea from the natural language processing (NLP) domain. Each of these embeddings has its own unique way of encoding the molecular structure and sequence information of a SMILES string.

### B.1  MORGAN FINGERPRINT

The Morgan fingerprint Nakajima & Nemoto (2021) is a circular fingerprint that encodes the presence of substructures within a molecule. Specifically, it generates a set of substructures of increasing diameter around each atom in the molecule and hashes these substructures into a fixed-length bit vector. The resulting fingerprint is a binary vector that encodes the presence or absence of these substructures.

### B.2  MACCS FINGERPRINT

The Molecular ACCess System (MACCS) fingerprint Keys (2005); Durant et al. (2002), on the other hand, is a binary fingerprint that encodes the presence or absence of predefined substructures. These substructures are defined based on the functional groups and ring systems commonly found in organic molecules. The resulting fingerprint is a binary vector that encodes the presence or absence of each substructure.

### B.3  $k$-MERS SPECTRUM

The $k$-mers-based spectrum is sequence-based embeddings that encode the frequencies of overlapping sub-sequences of length $k$ (also called n-Gram in the NLP domain). For $k$-mers, the sequence of characters in the SMILES string is divided into overlapping sub-sequences of length $k$. The frequency of each sub-sequence is then counted, and the resulting vector of frequencies is used as the embedding. Given an alphabet $\Sigma$ that corresponds to the set of unique characters within a SMILE string (i.e. #%()(+-.0123456789=@ABCDEFGHIKLMNOPRSTVWXYZ[\\]abcdefgilmnoprstuy/$), the length of $k$-mers spectrum is $|\Sigma|^k$ (where $|\Sigma| = 65$). In our experiments, we use $k = 3$, which is decided using the standard validation set approach.

### B.4 Minimizers Spectrum

Although minimizers have been well explored in the biology domain and proven to perform better than $k$-mers in preserving the information in biological sequences Zheng et al. (2020); Edgar (2021), their usage in the chemistry domain for SMILES strings analysis is not well known. Hence, we took the idea of minimizers from the biology domain and applied it to design embeddings for SMILES strings. Minimizers are a compact representation of substrings of a given sequence, such as SMILES strings. The minimizers (also called $m$-mers) are generated by selecting a fixed-length $k$-mer, known as a window, along the sequence and finding the lexicographically smallest (both in forward and reverse order) $k$-mer (based on ASCII values) in each window (where $m < k$), see Figure 2 for an example of the minimizer. To generate the minimizer-based spectrum, the minimizers are first sorted based on their position in the SMILES string. Then, each minimizer is assigned a unique index, and the frequency of each minimizer is counted. Based on the frequency count of $m$-mers within a SMILES string, we generate a spectrum where the length of the spectrum is $|\Sigma|^k$. By incorporating the use of minimizer-based spectra, we believe that it could be able to capture important information in the SMILES strings, such as functional group repeats and structural similarities, that may be missed by $k$-mers-based spectra. To generate the minimizers-based spectrum, we use $k = 5$ and $m = 3$, which is decided using the standard validation set approach.

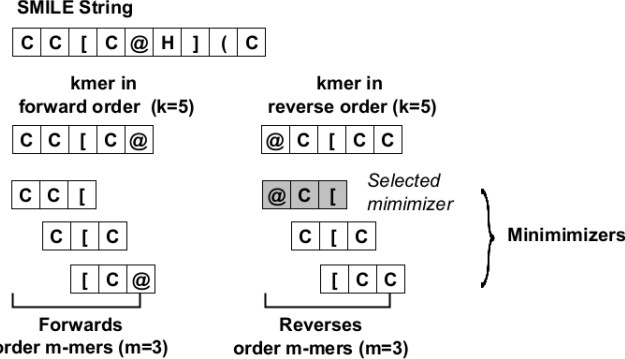

Figure 2: Example for generating minimizer from a sample SMILES string.

### B.5 Weighted $k$-mers Spectrum

Finally, we use a weighted version of $k$-mers that employs inverse document frequency (IDF) to assign weights to each $k$-mer within the spectrum Öztürk et al. (2020). This approach is based on the idea that frequent $k$-mers that appear in many SMILES strings are less informative than rare $k$-mers that appear in only a few SMILES strings. Therefore, we use inverse document frequency to down-weight the frequency of each $k$-mer based on the number of SMILES strings in which it appears. The resulting weighted $k$-mers-based embedding can better capture the unique features of each SMILES string, making it more effective for solubility prediction. For experiments, we selected $k = 3$, which is decided using the standard validation set approach. The pseudocode to compute weights for $k$-mers using IDF is given in Algorithm 1.

## C Experimental Evaluation

For the classification task, we use different linear and non-linear classifiers such as SVM, Naive Bayes (NB), Multi-Layer Perceptron (MLP), K Nearest Neighbors (KNN), Random Forest (RF), Logistic Regression (LR), and Decision Tree (DT). To evaluate the classification results, we report results for average accuracy, precision, recall, weighted F1, macro F1, ROC-AUC, and classifier training runtime. We split the data into $70 - 30\%$ random training and test sets and repeat the experiments 5 times to report average results. From the training data, we use $10\%$ data as a vali-

**Algorithm 1** $k$-mers weights computation using inverse document frequency (IDF).

```
 1: function COMPUTE_IDF_WEIGHTS(kmers_list)
 2:     num_samples ← length of kmers_list
 3:     idf_dict ← {}
 4:     for kmers in kmers_list do
 5:         for kmer in set(kmers) do
 6:             if kmer not in idf_dict then
 7:                 idf_dict[kmer] ← 0
 8:             end if
 9:             idf_dict[kmer] ← idf_dict[kmer] + 1
10:         end for
11:     end for
12:     for kmer, count in idf_dict do
13:         idf_dict[kmer] ← log(num_samples/count)
14:     end for
15:     return idf_dict
16: end function
```

dation set for hyperparameter tuning. Our code and pre-processed dataset are available online for reproducibility [1].

For regression analysis, we use models such as linear regression, ridge regression, lasso regression, random forest regression, and gradient boosting regression to predict solubility ALOGPS. Furthermore, we use multiple evaluation metrics, including mean squared error (MSE), mean absolute error (MAE), root mean squared error (RMSE), coefficient of determination ($R^2$), and explained variance score (EVS), to assess the performance of the models.

## C.1  DATASET STATISTICS

We use 6951 SMILES strings from the DrugBank dataset Shamay et al. (2018). For target labels, we use drug subtypes (total 188 unique subcategories) for classification tasks and solubility ALOGPS for regression analysis. The top 10 drug subcategories extracted from Food and Drug Administration (FDA) website [2] are given in Table 2. A sample SMILES string along with its attributes is shown in Table 3.

| Drug Subcategory | Count |
|---|---|
| Others | 6352 |
| Barbiturate [EPC] | 54 |
| Amide Local Anesthetic [EPC] | 53 |
| Non-Standardized Plant Allergenic Extract [EPC] | 30 |
| Sulfonylurea [EPC] | 17 |
| Corticosteroid [EPC] | 16 |
| Nonsteroidal Anti-inflammatory Drug [EPC] | 15 |
| Nucleoside Metabolic Inhibitor [EPC] | 11 |
| Nitroimidazole Antimicrobial [EPC] | 10 |
| Muscle Relaxant [EPC] | 10 |

Table 2: Top 10 drug subtypes extracted from Food and Drug Administration (FDA) website. The term EPC denotes "Established Pharmacologic Class".

## C.2  DATA VISUALIZATION

To visually evaluate whether different embedding methods are preserving the structure of the data, we use the t-distributed Stochastic Neighbour Embedding (t-SNE) Van der Maaten & Hinton (2008)

---

[1] https://github.com/sarwanpasha/Drug_Analysis

[2] https://www.fda.gov/

| SMILE String | Drug Name | Drug Subcategory | Solubility ALOGPS |
|---|---|---|---|
| CC[C@H](C)[C@H](NC(=O)[C@H](CCC(O)=O)NC(=O)[C@H](CCC(O)=O)NC(=O)[C@H](CC1=CC=CC=C1)NC(=O)[C@H](CC(O)=O)NC(=O)CNC(=O)[C@H](CC(N)=O)NC(=O)CNC(=O)CNC(=O)CNC(=O)CNC(=O)[C@@H]1CCCN1C(=O)[C@H](CCCNC(N)=N)NC(=O)[C@@H]1CCCN1C(=O)[C@H](N)CC1=CC=CC=C1)C(=O)N1CCC[C@H]1C(=O)N[C@@H](CCC(O)=O)C(=O)N[C@@H](CCC(O)=O)C(=O)N[C@@H](CC1=CC=C(O)C=C1)C(=O)N[C@@H](CC(C)C)C(O)=O | Bivalirudin | Anti-coagulant [EPC] | 0.0464 g/l |

Table 3: A Sample SMILES string with the drug name, drug subcategory, and Solubility ALOGPS values.

approach to design the 2-dimensional representation of the embeddings. The scatterplots of t-SNE using different embedding methods are shown in Figure 3. In general, we can observe some grouping for MACCS fingerprint while minimizers-based embedding tends to have fewer scattered points throughout the plot.

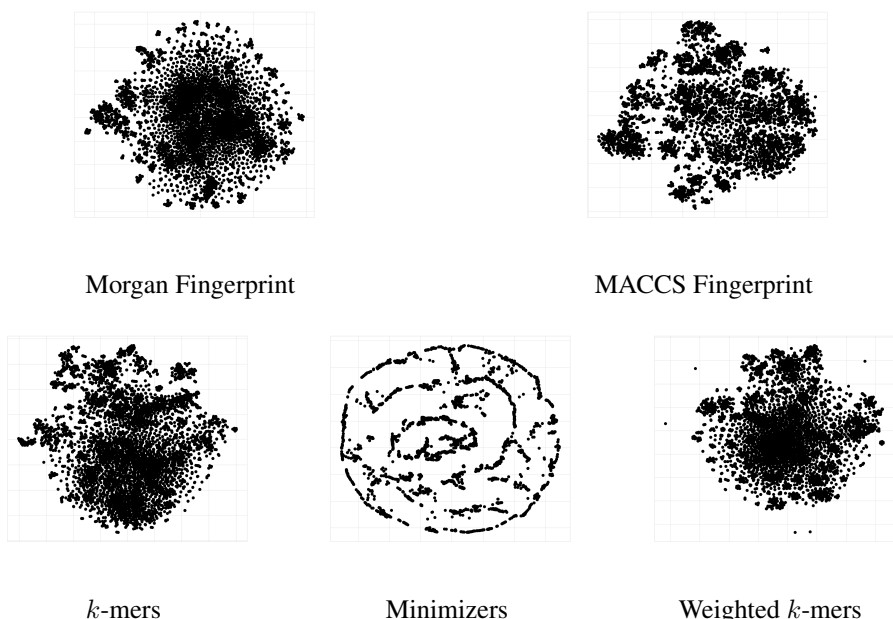

Morgan Fingerprint        MACCS Fingerprint

$k$-mers      Minimizers      Weighted $k$-mers

Figure 3: The t-SNE plots for different embedding methods.

# D  RESULTS

The regression results are shown in Table 4. From the results, it is clear that the Random Forest Regression model outperforms all other models for the MACCS Fingerprint with the lowest MAE of 17.8092, MSE of 3711.9790, and RMSE of 60.9260. It is also worth noting that the $R^2$ and EVS values are the highest for the Random Forest Regression model. This indicates that the Random Forest Regression model is the best-performing model for the MACCS Fingerprint representation.

| Embedding | Algo. | MAE ↓ | MSE ↓ | RMSE ↓ | $R^2$ ↑ | EVS ↑ |
|---|---|---|---|---|---|---|
| Morgan Fingerprint | Linear Regression | 63.2345 | 11601.2046 | 107.7088 | 0.3139 | 0.3143 |
| | Ridge Regression | 62.6110 | 11529.2733 | 107.3744 | 0.3182 | 0.3185 |
| | Lasso Regression | 53.4116 | 11043.7095 | 105.0890 | 0.3469 | 0.3474 |
| | Random Forest Regression | 24.0881 | 7722.9372 | 87.8802 | 0.5433 | 0.5439 |
| | Gradient Boosting Regression | 32.4982 | 8853.8418 | 94.0948 | 0.4764 | 0.4768 |
| MACCS Fingerprint | Linear Regression | 55.7719 | 11202.9967 | 105.8442 | 0.3375 | 0.3378 |
| | Ridge Regression | 55.5289 | 11167.1285 | 105.6746 | 0.3396 | 0.3399 |
| | Lasso Regression | 54.1349 | 11189.4825 | 105.7803 | 0.3383 | 0.3385 |
| | Random Forest Regression | **17.8092** | **3711.9790** | **60.9260** | **0.7804** | **0.7809** |
| | Gradient Boosting Regression | 31.4769 | 7308.5600 | 85.4901 | 0.5678 | 0.5678 |
| $k$-mers | Linear Regression | 8.3616e+10 | 4.6111e+23 | 6.7905e+11 | -2.72674e+19 | -2.72670e+19 |
| | Ridge Regression | 59.1402 | 12955.0398 | 113.8202 | 0.2339 | 0.2339 |
| | Lasso Regression | 51.7842 | 12608.1103 | 112.2858 | 0.2544 | 0.2545 |
| | Random Forest Regression | 23.2473 | 6073.5836 | 77.9331 | 0.6408 | 0.6420 |
| | Gradient Boosting Regression | 32.3582 | 8709.4397 | 93.3243 | 0.4849 | 0.4855 |
| Minimizers | Linear Regression | 50.5372 | 16914.9748 | 130.0575 | -0.00023 | 0.0 |
| | Ridge Regression | 50.5372 | 16914.9748 | 130.0575 | -0.00023 | 0.0 |
| | Lasso Regression | 50.5372 | 16914.9748 | 130.0575 | -0.00023 | 0.0 |
| | Random Forest Regression | 50.7730 | 16913.8831 | 130.0533 | -0.00017 | 0.0 |
| | Gradient Boosting Regression | 50.5372 | 16914.9748 | 130.0575 | -0.00023 | 0.0 |
| Weighted $k$-mers | Linear Regression | 1.3608e+11 | 1.6509e+24 | 1.2848e+12 | -9.7624e+19 | -9.7527e+19 |
| | Ridge Regression | 62.8535 | 13187.9852 | 114.8389 | 0.2201 | 0.2202 |
| | Lasso Regression | 55.5155 | 12241.4725 | 110.6411 | 0.2761 | 0.2762 |
| | Random Forest Regression | 24.0294 | 6224.7174 | 78.8968 | 0.6319 | 0.6330 |
| | Gradient Boosting Regression | 33.0856 | 9066.1662 | 95.2164 | 0.4638 | 0.4644 |

Table 4: Regression results for different models and evaluation metrics. The best values are shown in bold.

## E   DISCUSSION

Based on our classification and regression analysis, we can observe that different embedding methods and models perform differently for solubility and drug subcategory prediction tasks. For drug subcategory prediction, our study showed that the minimizers-based spectrum method outperformed the other embedding methods and classifiers in terms of average accuracy, precision, recall, and weighted F1 score. On the other hand, weighted $k$-mers performed better than other methods for Macro F1 and ROC-AUC. For solubility prediction, our study showed that the MACCS fingerprint with a random forest regression model performed better than all other embedding methods and regression models in terms of multiple evaluation metrics, including RMSE, MAE, and MSE. Moreover, the MAE value of weighted $k$-mers with random forest regression is comparable to that of the MACCS fingerprint. Similar is the case for RMSE, $R^2$, and EVS evaluation metrics. Although the traditional fingerprint methods were able to perform better for regression analysis, they failed to outperform the bioinformatics-based minimizers for the classification task.

## F   CONCLUSION

In conclusion, our study provides insights into the effectiveness of different embeddings, regression models, and classification models for drug solubility and drug subcategory prediction. This study's findings can be useful for future drug discovery research. Future work in this area could explore more sequence-based embedding methods, particularly those that are emerging from the bioinformatics domain (such as Spaced/gapped $k$-mers. Exploring graph-based approaches like graph neural networks (GNNs) for encoding molecular structures and predicting drug properties could be a promising direction. Another interesting direction could be to investigate the use of more advanced machine learning models like transformer-based models, which have shown great potential in natural language processing tasks, for drug discovery research. Finally, we will explore the use of multimodal learning approaches, combining molecular structure data with other sources of data, such as biological assays, to improve the accuracy of drug property prediction.

