# OpenReview forum: "When Biology has Chemistry: Solubility And Drug Subcategory Prediction using SMILES Strings"
_ICLR.cc/2023/TinyPapers — Submitted to Tiny Papers @ ICLR 2023_

### Official Review · Reviewer_Lp1p · 2023-03-27

**Confidence:** 4

**Summary Of Contributions:**

This paper investigates the effectiveness of different embedding methods on predicting (i) solubility ALOGPS and (ii) drug subcategories. The investigated embedding methods are Morgan fingerprint, MACCS fingerprint, k-mers, minimizer-based spectrum and an additional weighted version of k-mers that employs inverse document frequency.

**Rating:**

Great Start (GS): a submission which meets some of the reviewing criteria but has room for improvement

**Strengths And Weaknesses:**

**Strength**

Adopting an alternate embedding method from the bioinformatics domain to generate representations for SMILES strings is interesting and could be promising.

**Weaknesses**

The main contribution of the paper is not very obvious as the reader could be confused between the study of different embedding techniques and which technique is actually contributed by the authors

The experiments are not extensive enough. The authors should consider some benchmarking tasks that are popular for molecule property prediction classification and regression. Examples can be found in the paper: https://chemrxiv.org/engage/chemrxiv/article-details/628e5b4d5d948517f5ce6d72

There is no link to the code mentioned in the reproducibility section. The authors should consider using [anonymous GitHub](https://anonymous.4open.science/) for this purpose.

**Suggested Changes:**

Adding popular benchmark results would significantly improve the impact of the research which could be helpful in many other works relating to molecule property prediction. It will be great to see the performance of the embeddings on more recent models such as graph convolutional networks.

---

> ### Author Response · Authors · 2023-05-30
> **Response to The Reviewer's Comments**
>
> We thank the reviewer for their comments. Due to the tiny nature of the submission, we have not reported extensive experimental results (including comparisons with GCN) in the manuscript. However, exploring this area in more depth is definitely a potential future extension, which we will explore the extension of our paper.
>
> We have now included a GitHub link in the manuscript that contains the code for the embedding methods.

---

### Official Review · Reviewer_LBdq · 2023-03-30

**Confidence:** 4

**Summary Of Contributions:**

The paper focuses on predicting drug properties, specifically solubility and drug subcategories, using sequence-based embedding methods such as SMILES strings. The study investigates five types of embeddings: Morgan fingerprint, MACCS fingerprint, k-mers, weighted k-mers, and minimizer-based spectrum.

**Rating:**

Clear, Correct, and Reproducible (CCR): a submission which meets the reviewing criteria

**Strengths And Weaknesses:**

## Strengths

- The study investigates five different types of embeddings and evaluates their performance using multiple evaluation metrics, providing a comprehensive comparison of the different methods.

-  The study evaluates both classification and regression tasks, which are both important in drug discovery.

## Weaknesses

- The authors conducted their study in just on dataset of 6951 SMILES strings. An analysis to more and larger datasets would be beneficial.
The findings of the study could be limited to the specific characteristics of the dataset and may not be generalizable to other datasets or real-world drug molecules. Therefore, analyzing more and larger datasets could help to validate the findings and improve the generalizability of the results.

**Suggested Changes:**

See Weaknesses above

---

> ### Author Response · Authors · 2023-05-30
> **Response to The Reviewer's Comments**
>
> We thank the reviewer for their comments. Regarding the usage of a larger dataset, we can confirm that this is one of the potential future extensions of our work, where we will use larger datasets to analyze the proposed method for classification and regression. The current size of the data was selected based on the availability of the target labels. In the future, we will try to get labels for more SMILES strings.

---

### Comment · Area_Chair_2iAE · 2023-06-01
**This work meets the threshold for archival, contains the URM statement and is deanonymized**

---

### Meta-Review · Area_Chair_2iAE · 2023-04-08

**Recommendation:** Invite to archive
**Confidence:** 5

**Metareview:**

The authors evaluate various embedding methods together with various regression and classification models in solubility and drug subcategory prediction tasks with SMILES strings. The resulting findings may help practitioners select which combination of model and embedding method to use for their own tasks. The reviewers agree that the goal of mapping out this space is promising but point out that the limited experiments and lack of code make it difficult to evaluate correctness and reproducibility of the results. While the general idea is clear, the structure of the paper could be improved by shortening the abstract and intro in order to include more discussion of results and the conclusion in the main text instead of the appendix.

**Summary:**

The authors explore the combinatorial space of embedding methods, regression models and classification models for chemical structures. Reviewers agree this is promising but are concerned about limited evaluations.

**Reason For Not Giving A Higher Recommendation:**

Correctness and reproducibility are hard to evaluate as a result of the limited evaluations and no code being provided (though the authors did say they would provide it for final version). Clarity could also be improved by following the listed suggestions.

**Reason For Not Giving A Lower Recommendation:**

The goal is promising and promises to be impactful by helping practitioners choose the right combination of embedding methods and models.

---

### Decision · Program_Chairs · 2023-04-10

Invite to archive

---

> ### Author Response · Authors · 2023-05-30
> **Authors Response**
>
> We thank all the reviewers and the Meta reviewer for appreciating our work and giving us valuable feedback. We tried to incorporate the reviews and made a few changes in the writeup to make it more clear for readers. We have also included a GitHub link in the appendix that contains the code for the embedding methods discussed in the paper.
>
> Our paper is now de-anonymized and the URM statement is updated. We would like to mention that we wish to **opt-in** for archival.